

# Effect of *SlSAHH2* on metabolites in over-expressed and wild-type tomato fruit

Lu Yang[1], Yue Teng[1], Sijia Bu[1], Ben Ma[1], Shijia Guo[1], Mengxiao Liang[2] and Lifen Huang[3]

[1] College of Life Sciences, Anhui Normal University, Wuhu, Anhui, China
[2] College of Life Sciences, Nankai University, Tianjin, China
[3] Majorbio Bio-PharmTechnology Co. Ltd., Shanghai, China

## ABSTRACT

**Background:** Tomato (*Solanum lycopersicum*) is an annual or perennial herb that occupies an important position in daily agricultural production. It is an essential food crop for humans and its ripening process is regulated by a number of genes. S-adenosyl-l-homocysteine hydrolase (AdoHcyase, EC 3.3.1.1) is widespread in organisms and plays an important role in regulating biological methylation reactions. Previous studies have revealed that transgenic tomato that over-express *SlSAHH2* ripen earlier than the wild-type (WT). However, the differences in metabolites and the mechanisms driving how these differences affect the ripening cycle are unclear.

**Objective:** To investigate the effects of *SlSAHH2* on metabolites in over-expressed tomato and WT tomato.

**Methods:** *SlSAHH2* over-expressed tomato fruit (OE-5# and OE-6#) and WT tomato fruit at the breaker stage (Br) were selected for non-targeted metabolome analysis.

**Results:** A total of 733 metabolites were identified by mass spectrometry using the Kyoto Encyclopedia of Genes and Genomes (KEGG) database and the Human Metabolome database (HMDB). The metabolites were divided into 12 categories based on the superclass results and a comparison with the HMDB. The differences between the two databases were analyzed by PLS-DA. Based on a variable important in projection value >1 and $P < 0.05$, 103 differential metabolites were found between tomato variety OE-5# and WT and 63 differential metabolites were found between OE-6# and WT. These included dehydrotomatine, L-serine, and gallic acid amongst others. Many metabolites are associated with fruit ripening and eight common metabolites were found between the OE-5# *vs.* WT and OE-6# *vs.* WT comparison groups. The low L-tryptophan expression in OE-5# and OE-6# is consistent with previous reports that its content decreases with fruit ripening. A KEGG pathway enrichment analysis of the significantly different metabolites revealed that in the OE-5# and WT groups, up-regulated metabolites were enriched in 23 metabolic pathways and down-regulated metabolites were enriched in 11 metabolic pathways. In the OE-6# and WT groups, up-regulated metabolites were enriched in 29 pathways and down-regulated metabolites were enriched in six metabolic pathways. In addition, the differential metabolite changes in the L-serine to flavonoid transformation metabolic pathway also provide evidence that there is a phenotypic explanation for the changes in transgenic tomato.

**Discussion:** The metabolomic mechanism controlling *SlSAHH2* promotion of tomato fruit ripening has been further elucidated.

Corresponding author
Lu Yang, yanglu@ahnu.edu.cn

## INTRODUCTION

Tomato (*Solanum lycopersicum*), an annual or perennial herb, is an important agricultural crop (*Liu et al., 2021*). It is native to South America and has become one of the most important vegetable crops in China (*Su et al., 2021*; *Chen et al., 2009*). Tomato fruit are rich in nutrients such as lycopene, beta carotene, vitamin C, and reduced glutathione (*Huang et al., 2006*; *Li et al., 2020*). During fruit development, there are substantial changes in primary metabolites levels, including carbohydrates and acids. The fructose and glucose contents increase and the sucrose content falls as tomatoes ripen. In contrast, the malic, citric and quinic acid contents increase, but then decrease, with tartaric acid rising, as the fruit ripens (*Dou et al., 2022*). At the onset of ripening, flavonoids and carotenoids begin to accumulate (*Muir et al., 2001*). Tomato extracts reduce neoangiogenesis in breast tumors and have other potential anticancer effects (*Ma, Kang & Zhang, 2009*). It is a common model plant for studying fruit ripening (*Giovannoni, 2004*; *Inaba, 2007*; *Klee, 2004*) and some studies on the interactions among plant hormones, such as ethylene (ET) and gibberellin (GA), during tomato ripening have shown that GA treatment inhibited metabolite changes during ripening, while ET treatment promoted metabolite changes (*Park & Malka, 2022*). Knockdown of NAM ATAF1/2 CUC1/2 (NAC) transcription factors member *SlNAC9* has been shown to affect the metabolism of carotenoids and tomato fruit ripening (*Feng et al., 2023*). The glycine-rich RNA-binding protein RZ1A-Like (RZ1AL) also influences tomato fruit ripening by participating in the regulation of carotene biosynthesis and metabolism (*Li et al., 2022*).

S-adenosyl-L-homocysteine hydrolase (AdoHcyase, EC 3.3.1.1; SAHH) is present in a variety of cells and is closely related to DNA methylation because it catalyzes the reversible hydrolysis of S-adenosine homocysteine (SAH) into adenosine and homocysteine (Hcy) (*Turner et al., 2000*). In plants, SAHH regulates plant growth and development and pathogen defense. For example, interfering with *SAHH* gene expression in tobacco and *Arabidopsis thaliana*, led to dwarf plants with leaves that were wrinkled and dark green. They also had more lateral buds and aging was delayed (*Li et al., 2008*; *Mull, Ebbs & Bender, 2006*; *Rocha et al., 2005*; *Tanaka et al., 1997*). Inhibition of the *SAHH* coding gene *PvSAHH1* in switchgrass resulted in decreased SAH content, decreased lignin accumulation, and increased the enzymatic hydrolysis efficiency for cell wall polysaccharides (*Bai et al., 2018*). Furthermore, *SAHH* expression was significantly upregulated in potato leaves that had been infected with Phytophthora (*Arasimowicz-Jelonek et al., 2013*). Inhibition of *SAHH* in tobacco reduced viral replication and increased viral resistance (*Masuta et al., 1995*). In tomatoes, simultaneous suppression of the expression of three *SAHH* coding genes by viral-induced gene silencing resulted in severe stunting, drought resistance and resistance to *Pseudomonas syringae* (Pst DC3000) (*Li et al., 2015*). In addition, recent studies have shown that tomato *SAHH* also plays a role in fruit ripening. Microarray sequencing results showed that the expression of *SSN-U314915*

(corresponding to *SlSAHH2*) was significantly higher at the fruit breaker stage than at the mature green stage and sharply decreased after 1-methylcyclopropene treatment (*Yan et al., 2013*). In addition, overexpression of *SlSAHH2* in tomatoes promotes ethylene synthesis and accelerates fruit ripening (*Yang et al., 2017*).

Although transgenic tomatoes that overexpress *SAHH2* ripen earlier, the differences in metabolic components during ripening are not clear. Plant metabolomics is the qualitative and quantitative analysis of all metabolites in plant samples using a variety of highly sensitive instruments and common metabolomics techniques include liquid chromatography mass spectrometry (LC-MS) and gas chromatography-mass spectrometry (GC-MS) (*Chang & Wang, 2015*). Metabolomics is divided into untargeted metabolomics, targeted metabolomics, and broadly targeted metabolomics, where untargeted metabolomics is a comprehensive, unbiased analysis of metabolites in an organism. Untargeted metabolomic analyses are becoming popular as more information about testing substances becomes available (*Zhang & Chen, 2021*). Untargeted metabolism refers to metabolomic analyses that attempt to integrate biological life processes with end-product outcomes to provide a more complete understanding of the overall mechanism (*Guo et al., 2017*). In this study, two tomato lines that over-expressed *SAHH2* (*SAHH2*-OE) at the breaker stage (Br) were chosen for untargeted metabolome analysis (LC-MS) to further investigate the mechanism controlling the effect of *SlSAHH2* on tomato fruit metabolites.

# MATERIALS AND METHODS

## Plant materials and samples

The seeds from *SlSAHH2* over-expressed (OE-5# and OE-6#) and wild-type (WT) plants were placed in a shaker (220 rpm, 25 °C) and agitated until the seed hypocotyls had clearly developed. Then the sprouting seeds were planted in moist and fertile soil, covered with cling film, and grown at 25 °C with a photoperiod of 16 h of light and 8 h of darkness. The cling film was removed after the seeds had germinated. The seedlings were watered every 3 days and tomato fruits were collected at the breaker stage (Br). Fruits that were the same size were taken and immediately frozen in liquid nitrogen and stored at −80 °C. There were five biological replicates in the experiment.

## Experimental method

Each 50 mg sample was placed in a 2 mL centrifuge tube. A total of 400 μL of extraction solution (methanol:water = 4:1) was added to the tube, which also contained 0.02 mg/mL of internal standard (L-2-chlorophenyl alanine). The samples were then ground into a fine powder using a frozen tissue grinder at −10 °C for 6 min (50 Hz). The samples were cleaned using a temperature-controlled ultrasonic cleaner and freezing ultrasound at 5 °C for 30 min (40 kHz). After the ultrasound treatment, the samples were placed in a freezer at −20 °C for 30 min to precipitate the proteins. Then the samples were centrifuged at 4 °C for 15 min (13,000 *g*) and the supernatant was taken for LC-MS analysis.

## LC-MS analysis

The UHPLC-Q Exactive system and ultra-high performance liquid chromatography (UHPLC) tandem Fourier transform mass spectrometry (Thermo Fisher Scientific, Waltham, MA, USA) were used for determination. The chromatographic column was an Acquity UPLC HSS T3 (100 mm × 2.1 mm i.d., 1.8 μm; Waters Corp., Milford, MA, USA); mobile phase A, 95% water + 5% acetonitrile (0.1% formic acid); mobile phase B, 47.5% acetonitrile + 47.5% isopropanol + 5% water (0.1% formic acid); flow rate, 0.40 mL/min; sample size, 2 μL, and column temperature, 40 °C. Mass spectrum conditions: the positive and negative ion scanning ionization method was used to collect mass spectrum signals respectively; the scan type was 70–1,050; the sheath gas flow rate was 40 arb; the auxiliary gas flow rate was 10 arb, the heating temperature was 400 °C; capillary temperature, 320 °C; positive spray voltage, 3,500 V; and negative spray voltage, −2,800 V. Quality control samples (QCs) were prepared by mixing extracts from all the samples in an equal volume of liquid. The volume of each QC was the same as that of the samples and treated and tested in the same way as the samples that were used for analysis. In the instrumental analysis process, one QC sample was inserted into every 5–15 analysis sample to investigate the stability of the whole detection process.

## Data processing and identification of metabolites

The raw data were inputted into the Progenesis QI metabolomics processing software (Waters) after baseline filtering, peak recognition, integration, retention time correction, and peak alignment and the resulting data matrix included the retention time, mass charge ratio, and peak intensity. The MS and MS/MS mass spectral information was matched to the HMDB (http://www.hmdb.ca/), and the Metlin (https://metlin.scripps.edu/) and Majorbio databases. Data derived from the databases after screening were uploaded onto the Majorbio cloud platform (https://cloud.majorbio.com) for analysis. The spectrum peak intensity of a sample was normalized using a combined normalization method. At the same time, variables with a relative standard deviation greater than 30% of the QC samples were deleted and a log10 transformation was performed to obtain the final data matrix for subsequent analysis.

## Differential metabolite analysis

A principal component analysis (PCA) and orthogonal least partial squares discriminant analysis (OPLS-DA) were performed on the data matrix. In addition, Student's t-test and a difference multiple analysis were also performed. The differential metabolites were identified using the variable important in projection (VIP) obtained from OPLS-DA model and the $P$-value for Student's t test. Metabolites with a VIP > 1 and $P < 0.05$ were considered differential metabolites. A total metabolite classification map for the tomato fruit was obtained based on the superclass classification used by the HMDB.
The functional annotations of metabolic pathways were identified using KEGG to determine the pathways corresponding to the differential metabolites.

## RESULTS

### Main metabolites in WT and transgenic tomato fruit

The tomato fruit development and ripening periods include the immature green stage, mature green stage, breaker color stage (Br), orange color stage, red color stage, and overripe stage (Fig. 1A). The Br stage is often used as a sign for ripening (*Chen, 2020*). After analyzing tomato fruits at the Br stage using LC-MS, 733 metabolites were obtained from the primary and secondary mass spectrometry data (File S1) and 547 metabolites in 12 categories were identified using the HMDB (File S2). Among them, lipids and lipid molecules (41.13%) accounted for the largest proportion (File S2, Fig. 1B).

A total of 15 samples were divided into three groups, corresponding to WT, OE-5# and OE-6#, and five replicated samples were within the 95% confidence interval. Therefore, all the samples were valid and the three tomato lines were mostly separate from each other (Fig. 1C). The PLS-DA analysis showed that the main components of metabolism in the WT, OE-5# and OE-6# lines were different.

### Differentially accumulated primary metabolites in WT and transgenic fruit

The PLS-DA analysis showed that all five WT and OE-5# replicates were uniformly distributed in the 95% confidence rings with no abnormal samples, therefore each group of five replicates was a valid sample (Figs. 2A and 2B). The PLS-DA permutation test confirmed the results. The PLS-DA analysis showed that all five WT and OE-6# replicates were uniformly distributed in the 95% confidence rings with no abnormal samples; therefore each group of five replicates was a valid sample (Figs. 2C and 2D). The permutation test showed that the PLS-DA analysis was accurate. The distance between the experimental and control groups indicated that there were differences in metabolites between the two groups.

### Analysis of the different *SlSAHH2* metabolites in overexpressed tomato and WT tomato

The results showed that the variation trends for the differential metabolites between OE-5# *vs*. WT and OE-6# *vs*. WT were not identical. There were 103 different metabolites between OE-5# and WT, among which 58 red marked metabolites were up-regulated and 45 blue marked metabolites were down-regulated (Fig. 3A and File S3). Based on the screening conditions of a VIP > 1 and $P < 0.05$, 85 out of the 103 metabolites in OE-5# *vs*. WT group could be classified in the HMDB (File S4). The up-regulated metabolites included dehydrotomatine, L-serine and hesperetin, whereas the down-regulated metabolites included 4-ethylphenol, betaine aldehyde, and 3,4-dimethylstyrene. There were 63 different metabolites between OE-6# and WT, among which 58 red marked metabolites were up-regulated and five blue marked metabolites were down-regulated (Fig. 3B and File S5). There were 63 different metabolites between OE-6# and WT, among which 43 could be classified in the HMDB (File S6). The up-regulated metabolites included

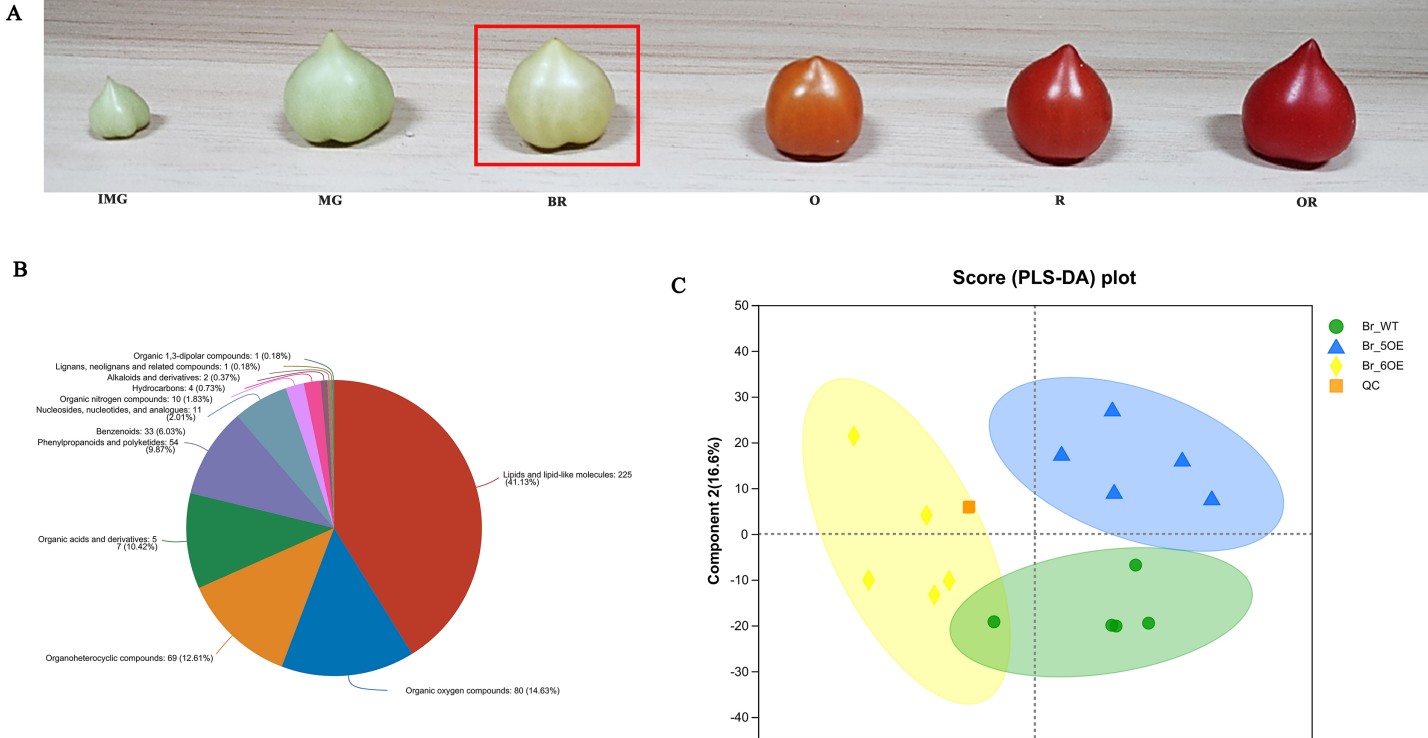

**Figure 1** **Overview of metabolic profiles of *SlSAHH2* overexpressed tomato and wild-type tomato.** (A) From the left to the right: the immature green stage (IMG), mature green stage (MG), breaker color stage (Br), orange color stage (O), red color stage (R) and overripe stage (OR). (B) Pie chart depicting the biochemical categories of the differential primary metabolites identified between *SlSAHH2* overexpressed tomato and wild-type tomato. (C) PLS-DA analysis of metabolites identified *SlSAHH2* overexpressed tomato and wild-type tomato. Equal volumes of the three varieties of tomato samples were mixed and used for quality control (MIX).

1,3-dicaffeoylquinic acid, kaempferol, and gallic acid. The down-regulated metabolites included L-tryptophan and pisumionoside.

Based on a VIP > 1 and *P* < 0.05, the number of compounds in the OE-5# *vs.* WT and OE-6# *vs.* WT comparison groups were 103 and 63, respectively. Among them, eight common metabolites were found in the two comparison groups, which were 2-aminomuconic acid semialdehyde, arginyl-aspartic acid, guanosine, cytosine, guanine, arginyl-glutamic acid, uracil, and L-tryptophan (Fig 3C). The fold changes in these eight common metabolites between the two transgenic lines and WT are shown in Fig. 3D.

## Analysis of the *SlSAHH2* over-expression tomato and WT enrichment pathways

The KEGG pathway enrichment analysis of the significantly different metabolites, the differential metabolites up-regulated in the OE-5# and WT groups were enriched in 23 metabolic pathways (File S7), including nine significant metabolic pathways (*P* < 0.05), namely ABC transporters, purine metabolism, pyrimidine metabolism, caffeine metabolism, valine, leucine and isoleucine biosynthesis, monobactam biosynthesis, glycine, serine and threonine metabolism, glycerophospholipid metabolism and

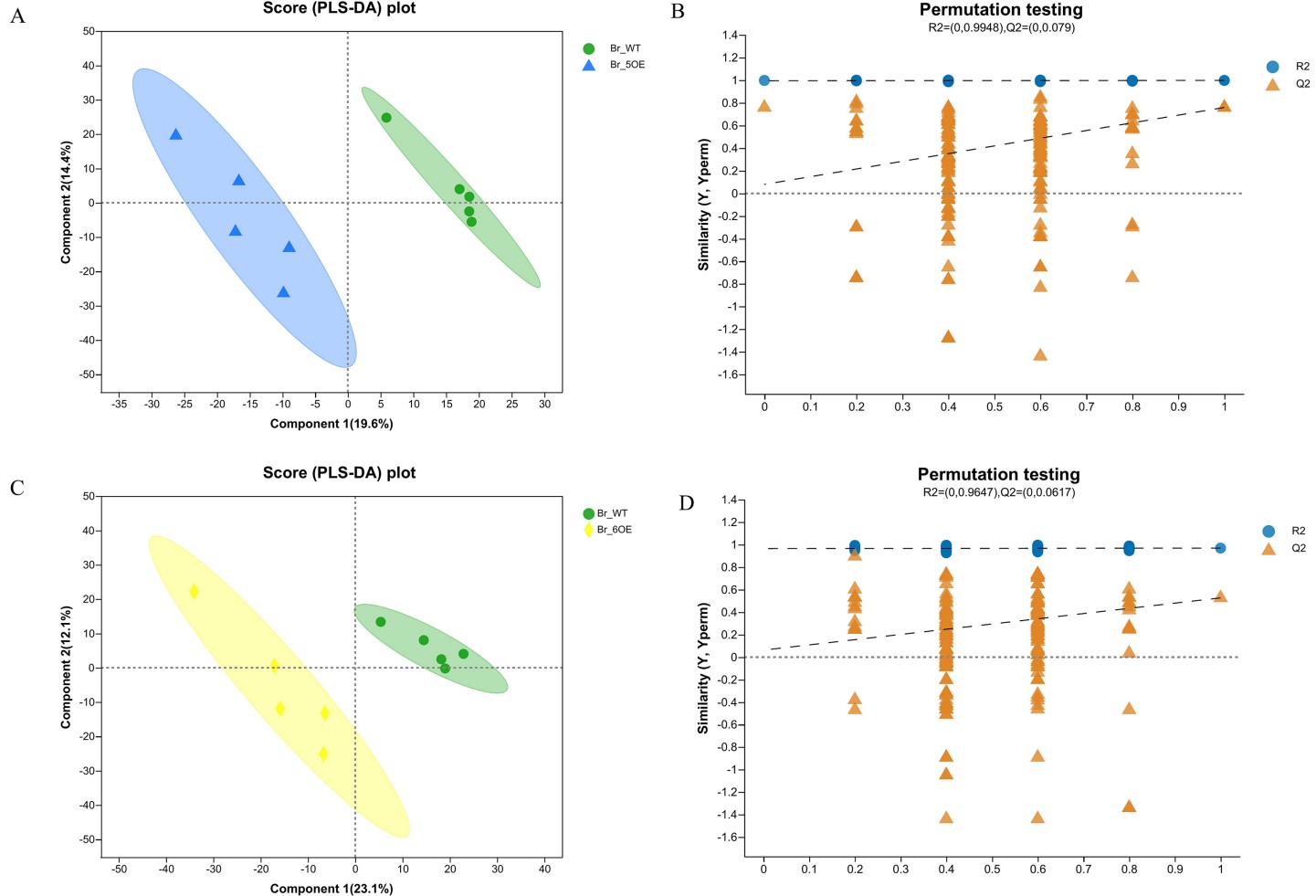

**Figure 2** **PLS-DA analysis of metabolites identified of *SlSAHH2* overexpressed tomato and wild-type tomato.** (A) Metabolites identified in OE-5# and WT *via* PLS-DA analysis. (B) The corresponding Permutation test of PLS-DA: R2 = (0, 0.9948), Q2 = (0, 0.079). (C) Metabolites identified in OE-6# and WT *via* PLS-DA analysis. (D) The corresponding Permutation test of PLS-DA: R2 = (0, 0.9647), Q2 = (0, 0.0617).

aminoacyl-tRNA biosynthesis (Fig. 4A). The differential metabolites that were down-regulated in the OE-5# and WT groups were enriched in 11 metabolic pathways (File S7), including six significant metabolic pathways ($P < 0.05$), namely glycine, serine and threonine metabolism, tyrosine metabolism, biosynthesis of various secondary metabolites-part 2, tryptophan metabolism, ubiquinone and other terpenoid-quinone biosynthesis, and phenylalanine, tyrosine, and tryptophan biosynthesis (Fig. 4B). The differential metabolites that were up-regulated in the OE-6# and WT groups were enriched in 29 metabolic pathways (File S8), including six significant metabolic pathways ($P < 0.05$), namely purine metabolism, aminoacyl-tRNA biosynthesis, pyrimidine metabolism, ABC transporters, beta-alanine metabolism and tropane, piperidine and pyridine alkaloid biosynthesis (Fig. 5A). The differential metabolites that were down-regulated in the OE-6# and WT groups were enriched in six metabolic pathways (File S8) and all six were significant metabolic pathways ($P < 0.05$), namely phenylalanine, tyrosine, and tryptophan

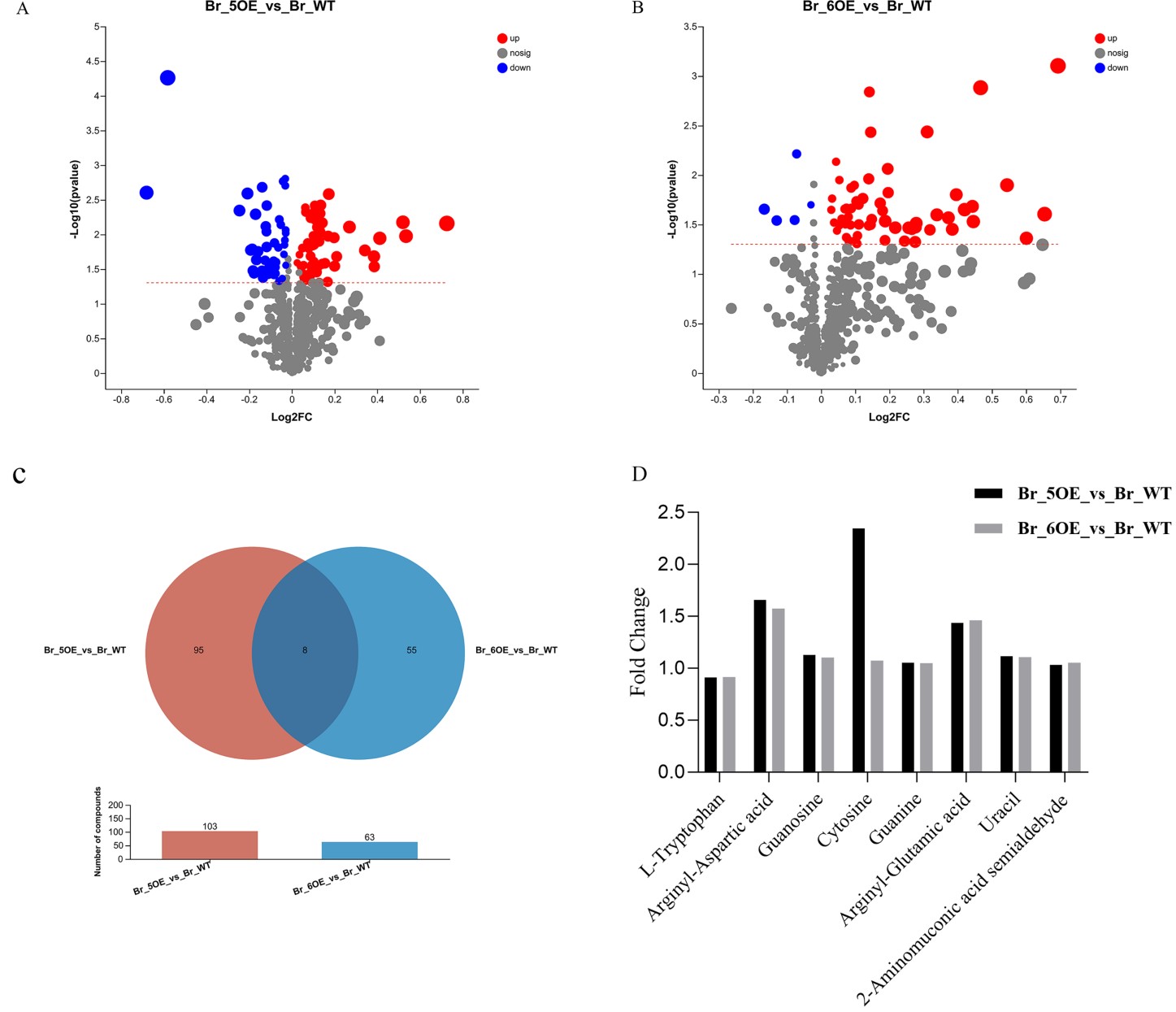

**Figure 3** **Volcano plot and Venn diagram of the 733 metabolites identified.** (A) Differential metabolites were defined as metabolites with VIP > 1 and *P* < 0.05 in OE-5# compared to WT. (B) Differential metabolites were defined as metabolites with VIP > 1 and *P* < 0.05 in OE-6# compared to WT. (C) Based on a VIP > 1 and *P* < 0.05, the number of common metabolites in the OE-5# *vs*. WT and OE-6# *vs*. WT groups was eight. (D) The fold changes in the these eight common metabolites between the two transgenic lines and WT.

biosynthesis, glycine, serine, and threonine metabolism, aminoacyl-tRNA biosynthesis; glucosinolate biosynthesis, biosynthesis of various secondary metabolites-part 2, and tryptophan metabolism (Fig. 5B).

## DISCUSSION

Metabolomics is a broad, sensitive, and practical approach and the analysis of plant phenotypes by differential metabolites is a commonly used method (*Patel et al., 2021*).

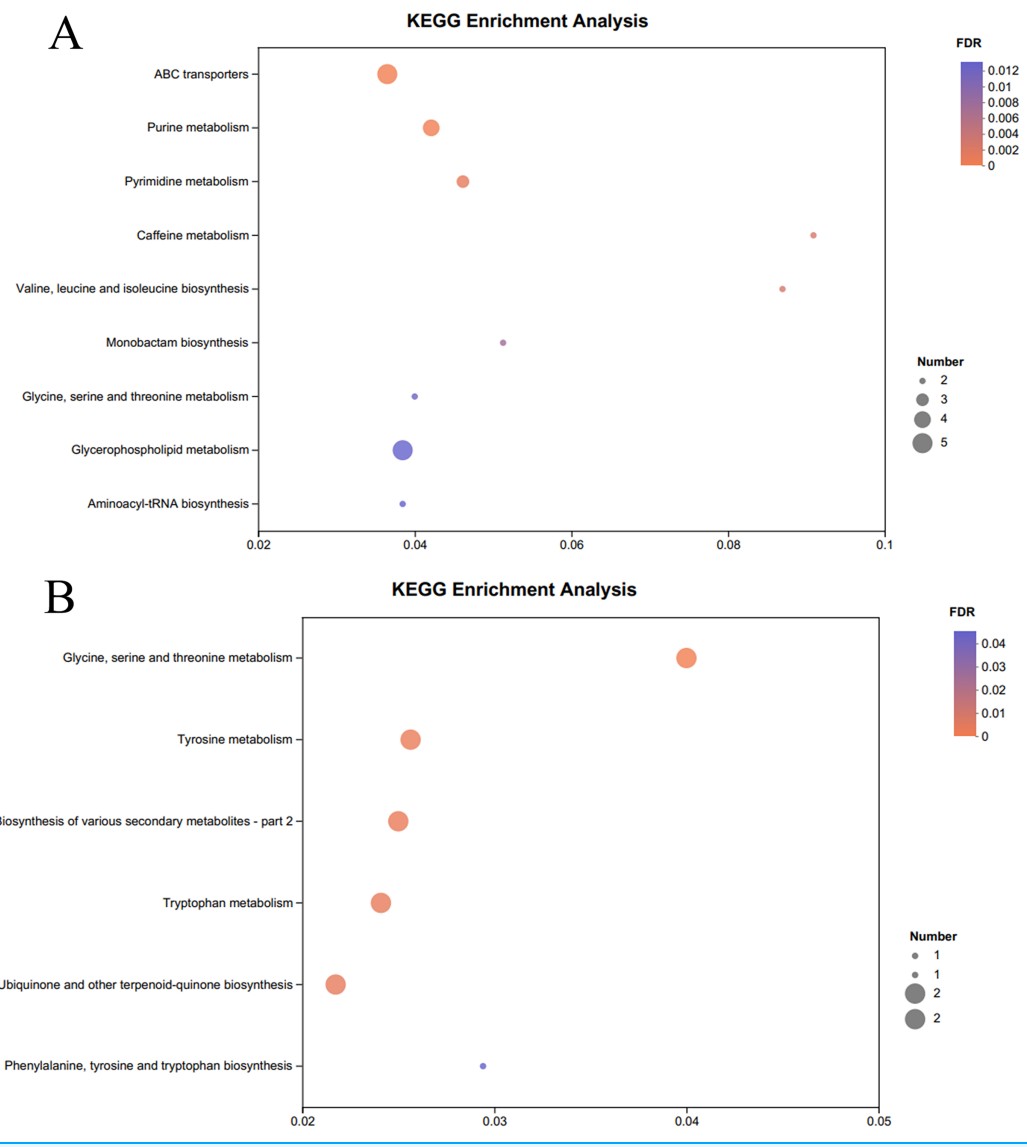

**Figure 4** **A total of 733 metabolites were identified by bubble diagrams.** (A) OE-5# and WT up-regulated metabolites enriched by KEGG into the pathway shown by bubble diagrams (VIP > 1 and *P* < 0.05). (B) OE-5# and WT down-regulated metabolites enriched by KEGG into the pathway shown by bubble diagrams (VIP > 1 and *P* < 0.05).

*Kissoudis et al. (2015)* studied the changes to the metabolites produced in tobacco plants that overexpressed glutathione S-transferases under salt stress. Protective metabolites, such as proline and trehalosaccharide, in the transgenic plants were present at higher concentrations under salt stress and the results showed that GmGSTU4 contributed to salt stress tolerance (*Kissoudis et al., 2015*). In this study, different metabolites in tomato plants that overexpressed *SAHH2* compared to the WT tomatoes were compared to explore the early ripening mechanism in tomatoes that overexpressed *SAHH2*. The metabolites produced by transgenic and WT plants were screened by LC-MS non-targeted liquid chromatography metabolomics. A comparison of the superclass results using HMDB showed that the metabolites were divided into 12 classes. A PLS-DA analysis and other

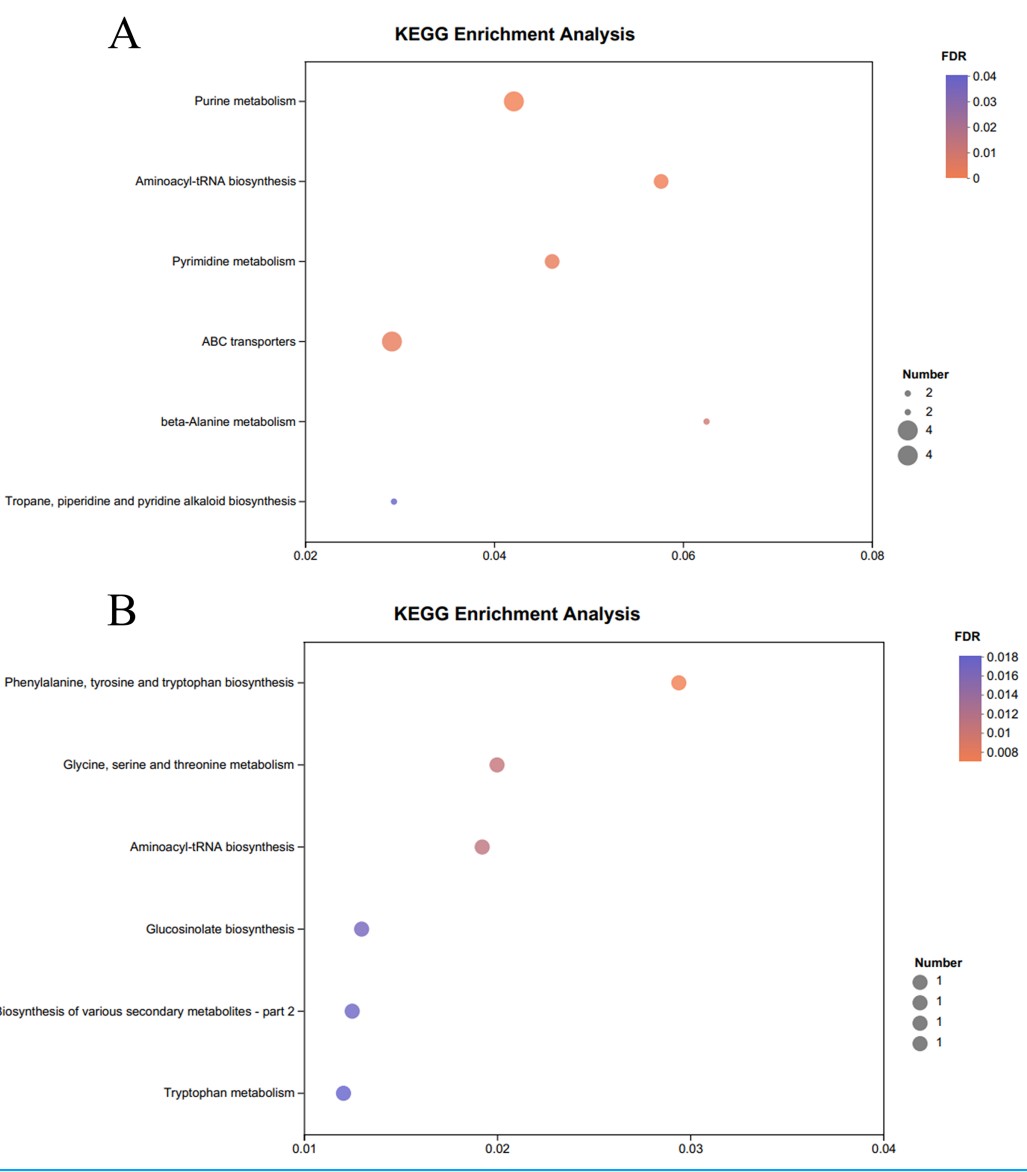

**Figure 5** **A total of 733 metabolites were identified by bubble diagrams.** (A) OE-6# and WT up-regulated metabolites enriched by KEGG into the pathway shown by bubble diagrams (VIP > 1 and $P < 0.05$) (B) OE-6# and WT down-regulated metabolites enriched by KEGG into the pathway shown by bubble diagrams (VIP > 1 and $P < 0.05$).               

methods were used to analyze the differences between transgenic and WT plants. The upregulated metabolites between OE-5# and WT were mainly dehydrotomatine and L-serine. Dehydrotomatine and L-serine play key roles in tomato fruit development and ripening. Dehydrotomatine accumulates in tomato leaves, tomato flowers and ripe green fruits and has a defense effect against bacteria, fungi and insects (*Lee et al., 2019*). Serine also increases during fruit ripening due to glycolysis (*Lee et al., 2010*). The important upregulated metabolites between OE-6# and WT were 1,3-dicaffeoylquinic acid, kaempferol, and gallic acid. Kaempferol, and gallic acid are involved in maturation. For example, kaempferol contents have been shown to be highest at the ripening phase

during grape development (*Fang, Tang & Huang, 2013*) and gallic acid increased from the pink to red stages during mulberry ripening (*Saracoglu, 2018*). The kaempferol compounds begin to accumulate in tomatoes during the color turning stage and the flavonoid content reaches a maximum at ripening (*Le Gall et al., 2003*). Flavonoids act as ROS to scavenge antioxidants in type VI trichomes and flavonoid deficiency disrupts redox homeostasis and terpenoid biosynthesis in the glandular trichomes of tomato (*Sugimoto et al., 2022*). In tomato fruits, the low levels of flavonoid related transcripts in the flesh means that flavonoids mainly accumulate in the peel (*Dal Cin et al., 2011*; *Zhang et al., 2015*).

In a previous study, LC-MS/MS was used to analyze the differential metabolites between tomato mature green and red ripe fruit and the results showed that there were 652 differential metabolites, including 361 up-regulated and 219 down-regulated metabolites. These metabolites may contribute to flavor accumulation in tomato (*Zuo et al., 2020*). Among the 652 differential metabolites, three (guanosine, cytosine and L-tryptophan) out of the eight common metabolites in the OE-5# and WT and OE-6# and WT comparison groups were also identified in this study (*Zuo et al., 2020*, Fig. 3C). This correlation explained the differences in fruit ripening period between transgenic and WT tomatoes at the metabolite level. Tryptophan has multiple functions in plants. For example, the first step in serotonin synthesis is the conversion of tryptophan into tryptamine by tryptophan decarboxylase (*Islam et al., 2016*). Overexpression of the tryptophan decarboxylase gene (*SlTDC1*) in tomato revealed the physiological roles of tryptophan decarboxylase and indicated the role of serotonin in ripening (*Tsunoda et al., 2021*). Furthermore, changes to the pathways related to amino acid biosynthesis of L-tryptophan, L-valine, L-alanine, L-glutamic acid, L-leucine, L-histidine occurred in fruit at the mature green and red ripe stages (*Zuo et al., 2020*). *Zuo et al. (2020)* also reported that the tryptophan content in the red ripe fruit was downregulated compared to mature green fruit. In this study, the tryptophan content in transgenic fruit (earlier ripening) was also downregulated, which was consistent with *Zuo et al. (2020)* (Fig. 3D).

In the cysteine and methionine metabolism pathway, SAH is hydrolyzed by SAHH to generate Hcy, which suggests that the Hcy in tomatoes overexpressing *SAHH2* is probably up-regulated. Generally, L-serine and Hcy produce L-cystathionine using cystathionine β-synthase (*Xu, Hu & Wang, 2017*). L-serine in OE-5# and OE-6# was up-regulated compared to the WT (Fig 6), which suggests that L-cystathionine was also up-regulated. At the same time, under the action of γ-lyase, L-cystathionine produces L-homocysteine, which means that L-homocysteine in transgenic fruit may be up-regulated compared to the WT. Furthermore, L-methionine is produced from L-homocysteine using methionine synthase, therefore L-methionine in transgenic fruit may also increase (*Shen, 2019*). When combined with the trans-methylation pathway for methionine, S-adenosyl-L-methionine can be produced from L-methionine using methionine adenosine transferase. S-adenosyl-L-methionine can accelerate the production of 1-aminocyclopropane-1-carboxylic acid and S-methyl-5′-thioadenosine using S-adenosine-l-methionine methionine adenosine lyase (*Avila et al., 2004*; *Tang et al., 2008*).

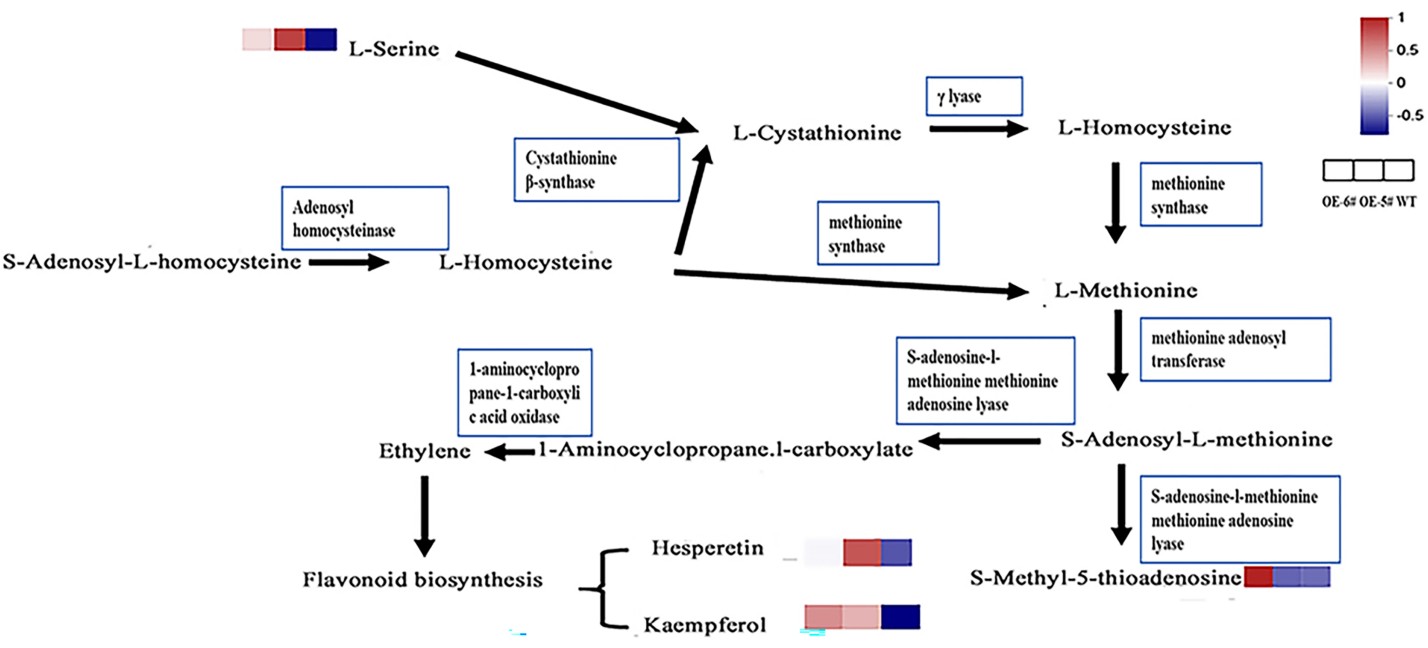

**Figure 6   The inferred metabolic mechanism of *SlSAHH2* promoting fruit ripening.**     

The contents of the terminal chemical compounds hesperetin, kaempfrerol, and S-methyl-5-thioadenosine were all higher in the transgenic tomato fruit than in the WT fruit (Fig 6). 1-aminocyclopropane-1-carboxylate (ACC) is the precursor of ethylene synthesis and the up-regulation of ACC can reflect increases in ethylene in fruit (*Mou et al., 2020*; *Xie, Ying & Ying, 2009*). It has also been shown that ethylene promotes flavonoid synthesis (*Ni et al., 2020*). The inferred increase in S-adenosyl-L-methionine content and the reported high ethylene level in transgenic fruit (*Yang et al., 2017*) further clarify the metabolic mechanism underlying *SlSAHH2* promotion of fruit ripening.

## CONCLUSIONS

In this study, LC-MS non-targeted liquid chromatography metabolomics was used to screen the metabolites produced by *SlSAHH2*-overexpressed transgenic and WT tomato fruit at the Br stage. PLS-DA was then used to analyze whether there were any differences between the tomato lines. Based on the screening conditions: VIP > 1 and $P < 0.05$, 103 differential metabolites between tomato variety OE-5# and WT and 63 differential metabolites between OE-6# and WT were identified. Then, corresponding screening was conducted and the metabolites were labeled using the HMDB and KEGG database to obtain a total metabolite classification map for tomato. The L-serine, kaempferol, hesperidin, and other ripening related substances levels in overexpressed tomato were higher than those in WT tomato. The L-tryptophan content in the transgenic lines was lower than in the WT fruits, which was consistent with previous reports that its content decreased with fruit ripening. The results indicated possible reasons for the different ripening periods between transgenic and WT fruit from the metabolites and components

perspectives. They also provide further information about the role played by *SlSAHH2* in tomato fruit ripening.

## ACKNOWLEDGEMENTS

The authors would like to thank Tong Zhou (Shandong University) for his technical support.

### Funding

This research was supported by the National Natural Science Foundation of China (32302166) and the 17th Huo Yingdong Education Fund Project (171022). The funders had no role in study design, data collection and analysis, decision to publish, or preparation of the manuscript.

### Grant Disclosures

The following grant information was disclosed by the authors:
National Natural Science Foundation of China: 32302166.
17th Huo Yingdong Education Fund: 171022.

### Competing Interests

Huang Lifen is employed by Majorbio Bio-PharmTechnology Co. Ltd. (Shanghai).
The authors declare that they have no competing interests.

### Author Contributions

- Lu Yang conceived and designed the experiments, authored or reviewed drafts of the article, and approved the final draft.
- Yue Teng analyzed the data, authored or reviewed drafts of the article, and approved the final draft.
- Sijia Bu conceived and designed the experiments, performed the experiments, analyzed the data, prepared figures and/or tables, authored or reviewed drafts of the article, and approved the final draft.
- Ben Ma performed the experiments, analyzed the data, prepared figures and/or tables, and approved the final draft.
- Shijia Guo performed the experiments, analyzed the data, prepared figures and/or tables, and approved the final draft.
- Mengxiao Liang analyzed the data, prepared figures and/or tables, and approved the final draft.
- Lifen Huang performed the experiments, prepared figures and/or tables, and approved the final draft.

### Data Availability

The data is available at figshare: Yang, Lu; Teng, Yue (2023). Tomato metabolome data. figshare. Dataset. https://doi.org/10.6084/m9.figshare.24543055.v2.

## Supplemental Information

Supplemental information for this article can be found online at http://dx.doi.org/10.7717/peerj.17466#supplemental-information.

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
