# Peer review of "Effect of SlSAHH2 on metabolites in over-expressed and wild-type tomato fruit"

_PeerJ, doi:10.7717/peerj.17466_

## Round 0.1 · original submission · Major Revisions

Dear Authors

The manuscript cannot be accepted for publication in its current form. The manuscript needs substantial revision to meet the standards of the journal. The authors are invited to revise the paper considering all the suggestions made by both reviewers, including reviewer 5 who recommended rejecting the manuscript. Please note that requested changes are required for publication.

With Thanks

**Language Note:** The review process has identified that the English language must be improved. PeerJ can provide language editing services - please contact us at [email protected] for pricing (be sure to provide your manuscript number and title). Alternatively, you should make your own arrangements to improve the language quality and provide details in your response letter. – PeerJ Staff

·

Basic reporting

The paper investigated the effect of SlSAHH2 on metabolites between over-expressed tomato and wild-type tomatoes.
The paper is relatively well written, except for some parts (see below) that require clarification. Since I did not receive the manuscript in a line numbering format, I couldn’t generate a separate report. Most of my comments/inquiries can be fixed and the authors should have answers to them. Ultimately, the paper meets the requirements of scientific publications in terms of format and objective, so I recommend publishing after some minor amendments

Experimental design

Good experimental design that fulfilled the objectives of the study

Validity of the findings

The paper reported that 733 metabolites were obtained which were divided into 12 categories. The contents of serine, tryptophan, kaempferol, hesperidin and other ripening related substances in overexpressed tomato were higher than those in wild type tomato. According to the results, genetically modified tomatoes may ripen more easily than wild-type tomatoes. The results showed that SlSAHH2 had a certain effect on tomato fruit ripening, which provided a certain reference value for the subsequent research on SlSAHH2.

Reviewer 2 ·

Basic reporting

This MS studied the role of SAHH2 on metabolites between over-expressed tomato and wild type tomato. Below are the comments.
The abstract did not provide enough details of different metabolites in the overexpressing lines. At least the most significant changes should be shown. Besides, the figures also did not provide enough information, and only overall analysis shown.
The phenotype of overexpressing lines and WT fruits during fruit ripening should be shown in figure 1 which at present only WT.
What the differences between OE-5 and OE-6 at the expression of SAHH2. The author should provide more details.
Figure 4, the up-regulated and down-regulated metabolites should be analyzed separately.

Experimental design

no comment

Validity of the findings

no comment

Additional comments

no comment

Reviewer 3 ·

Basic reporting

Yang et al. reported metabolite profiles between two tomato lines overexpressing SlSAHH2 and a wild type fruit. Metabolite profiling was performed using LC-MS and the authors were able to find serine, tryptophan, kaempferol, and hesperidin that were up-accumulated in the overexpression lines. Although the manuscript looks interesting in term of showing metabolic changes under the effect of SlSAHH2, I found out only metabolome data it is very difficult to draw strong conclusions from their observations.

Here are some other comments;
Comment on language: The English language should be improved. There are some grammatical errors, such as:

Line 59; not 'thalian'
Line 101-102: ground samples with a grinder…; this should be: Samples were ground into a fine powder using….

Line 101: extraction solution contains…..; this should be: contained

Line 104: Centrifuge at….; this should be: Samples were then centrifuged at……

Experimental design

The design was fine but there are comments on M&M
Line 94: How did you plant your seeds? Please provide more details, such as light/dark cycle, humidity, etc.

Line 96: Five biological replicates were used in this study. Please define the biological replicates. Do you mean five fruits from five different plants?

Line 98: metabolite extraction?

Line 99: Each sample weighed…. What is the definition of “sample”?

Line 100: The extraction is V/V?

Line 102: tissue grinder at -10? What is the grinder specification?

Line 104: please correct the centrifugation speed (either rpm or x g).

Line 107: Please define the abbreviations when used for the first time.

Section 2.4: Is it data processing or statistical analysis?

Validity of the findings

The authors detected many metabolites but the question is how those metabolites are related to the function of SlSHH2. Only a few metabolites were mapped to the pathway shown in Fig. 5 and this makes it difficult to propose a metabolic mechanism of SlSAHH2 to promote fruit ripening. Some other target metabolites in the pathway must be analyzed before the conclusion can be drawn. Or the expressions of those genes related to the pathway must be analyzed.

For PLS-DA, authors should provide Variable in projection (VIP) scores for the top metabolites contributed to the clustering (separation).

For defining the differential metabolites, the authors used the P<0.05. How did you perform the statistical analysis of the metabolites? Which approach did you use to compare between the two groups?

What is the definition of VIP>1 in Figure 3?

Line 177: and so on? What does it mean? I am not sure if hesperetin can be found in tomato fruit.

Did OE-5 vs wild share the same differential metabolites compared to OE-6 vs wild? In other words, did OE-5 and OE-6 lines harbour the same metabolic shifts which resulted in accelerated ripening?

Line 229-242: authors used a lot of speculations in the up-regulated metabolites. At least, some solid evidence should be provided for some metabolites indicated in Fig. 5.
The major concern of this study: if the over expression of SAAH2 results in faster fruit ripening via enhancing ethylene biosynthesis (as authors claimed), confirmation is needed. Ethylene measurements should be performed and/or RT-qPCR for confirming the higher expressions of ACS and ACO in transgenic lines.

How many paralogues of SAAH are available in the tomato genome? Why did the authors focus on SAAH2?

Regarding the higher contents of flavonoids in the transgenic lines, more discussion is needed (possible role is stress tolerance).

Among the differentially accumulated metabolites, amino acids such as tryptophan were highlighted by the authors (which I agree). However, the potential roles of these amino acids during climacteric fruit ripening are needed to be discussed, such as their involvements as precursor or intermediate compounds in different biosynthetic pathways.

Another major concern about this study: How did the authors perform the statistical analysis? the authors only mentioned the p value (that is fine), but details about the approach used to analyze the data should be provided.

Additional comments

Line 224-228: Please provide some references of climacteric fruits since the focus of this study is on climacteric ripening.

Please double check the manuscript for some typo errors, such as line 260: he results…….

·

Basic reporting

The manuscript needs to be consistent in its grammar: please leave a space before opening parenthesis (examples: line 40, line 42, line 44, etc).
Line 63: leave a space after punctuation and before “In terms”.
Line 94: don’t leave a space after “25”.
Line 100: don’t leave a space between “:” and “water”.

Line 59: Arabidopsis thaliana is misspelled.

Line 69: Pseudomonas syringae needs to be italicized.

The manuscript should be written in a clear and diplomatic manner. For instance, instead of using “so on” in lines 177 and 182, a more appropriate phrase should be used.

Line 238: replace “increases” with “increase”.

Line 253: add the word “tomatoes” at the end of the sentence.

Line 260: replace “he” with “The”.

Experimental design

No objections.

Validity of the findings

Line 206: Please further elaborate on the statement “The analysis of plant phenotypes by differential metabolites is a good method”: what is implied by “good method”?

Additional comments

I would like to highlight the brief yet focused introduction that was drafted. Additionally, it may be advantageous to revisit the key concepts in the conclusion to further underscore the significance of the findings.

Reviewer 5 ·

Basic reporting

In this manuscript, Yang and Colleagues present an analysis of the metabolomic effects of the overexpression of the SlSAHH2 (S-Adenosyl-l-Homocysteine hydrolase) in tomato fruits. The authors use a popular approach to understand the effect of specific genes in the regulation of specialized metabolism. The research shows potentially interesting results but unfortunately is very descriptive and fails to provide the necessary background and implications of the study. As such, I recommend the rejection of this manuscript. I believe that the research can eventually be published in a more specialized journal.

In general, the manuscript is not written clearly and has multiple terms that are not described properly (“over-expressed tomato”, PLSDA, VIP, P) and the background of the study is missing (for instance the references about the mutant lines). Additionally, the scientific problem, objectives and rationale behind the methodological approach are not presented in sufficient detail. For instance, some questions that need to be addressed are, what is the importance of studying the SAHH gene and its function in fruit ripening? why do you use the breaker stage (Br)? Although the interpretation of results seems correct, it is relatively superficial and does not address the implications of observed results.

Experimental design

This is a relatively descriptive study based on a small experiment.

The scope of the study is limited and there is no clear explanation of why the methods were used and how the results were interpreted.

For example, important details are missing about how the mutants were produced and maintained. Additionally there is no clear explanation on why this study was only performed at the Br stage.

Validity of the findings

no comment

---

## Round 0.2 · Minor Revisions

Dear Authors
The manuscript needs a minor revision to be reconsidered for publication. The authors are invited to revise the paper considering all the suggestions made by the reviewers.
With Thanks

Reviewer 2 ·

Basic reporting

Figure 4 is not clear.

Experimental design

no comments

Validity of the findings

no comments

Additional comments

no comments

Reviewer 3 ·

Basic reporting

The authors have extensively revised the manuscript and responded all my comments. However, English is still not acceptable in this current version, e.g. line 19 ' over-expressed of SlSAHH2. line 20 what is metabolomics mechanisms, 'fruit' are both plural and single form. 'Fruits' is used when many kinds of fruit are mentioned. I strongly suggest that this manuscript must be read and corrected by a native speaker before it can be accepted.

Experimental design

no comment

Validity of the findings

no comment

Additional comments

no comment

·

Basic reporting

No comment

Experimental design

No comment

Validity of the findings

No comment

---

## Round 0.3 · accepted · Accept

Dear Authors,

I am pleased to inform you that the manuscript has improved after the last revision round and can be accepted for publication.

Congratulations on accepting your manuscript, and thank you for your interest in submitting your work to PeerJ.

With Thanks